# Finite Plane Strain Bending under Tension of Isotropic and Kinematic Hardening Sheets

**DOI:** 10.3390/ma14051166

**Published:** 2021-03-02

**Authors:** Stanislav Strashnov, Sergei Alexandrov, Lihui Lang

**Affiliations:** 1General Education Courses Department, Рeoples’ Friendship University of Russia (RUDN University), 6 Miklukho-Maklaya Street, 117198 Moscow, Russia; shtrafnoy@gmail.com; 2School of Mechanical Engineering and Automation, Beihang University, 37 Xueyuan Road, Beijing 100191, China; sergei_alexandrov@spartak.ru; 3Faculty of Materials Science and Metallurgy Engineering, Federal State Autonomous Educational Institution of Higher Education “South Ural State University (National Research University)”, 76 Lenin Prospekt, 454080 Chelyabinsk, Russia

**Keywords:** bending under tension, large strain, isotropic hardening, kinematic hardening, semianalytic solution

## Abstract

The present paper provides a semianalytic solution for finite plane strain bending under tension of an incompressible elastic/plastic sheet using a material model that combines isotropic and kinematic hardening. A numerical treatment is only necessary to solve transcendental equations and evaluate ordinary integrals. An arbitrary function of the equivalent plastic strain controls isotropic hardening, and Prager’s law describes kinematic hardening. In general, the sheet consists of one elastic and two plastic regions. The solution is valid if the size of each plastic region increases. Parameters involved in the constitutive equations determine which of the plastic regions reaches its maximum size. The thickness of the elastic region is quite narrow when the present solution breaks down. Elastic unloading is also considered. A numerical example illustrates the general solution assuming that the tensile force is given, including pure bending as a particular case. This numerical solution demonstrates a significant effect of the parameter involved in Prager’s law on the bending moment and the distribution of stresses at loading, but a small effect on the distribution of residual stresses after unloading. This parameter also affects the range of validity of the solution that predicts purely elastic unloading.

## 1. Introduction

Finite plane strain bending with and without tension is one of the classical problems of plasticity. All sheet-metal processes incorporate some bending, which demonstrates the practical importance of this problem. A brief overview of such processes was presented in [1]. Moreover, the process of bending is often used as a test for determining material properties, which requires a theoretical description of the process. In particular, cyclic bending tests were employed in [2,3]. A four-point bending test was carried out in [4] for investigating the large-strain elastic constitutive behavior of thin unidirectional composites. Anisotropic hardening of metal sheets was studied in [5]. An in-plane bending test was used in [6] for determining the uniaxial large strain work-hardening behavior of high-strength steel. A pure bending test was employed in [7] to study of the strength differential effect of DP98 steel sheets.

The first rigorous rigid plastic solution was found in [8] for the model of isotropic perfectly plastic material. The work in [9] extended this solution for bending with no tension to rigid plastic, work-hardening material. However, this solution assumes an artificial strain distribution between the current neutral line and the original center fiber. The study in [10] further extended the solution for pure plane strain bending given in [8] to rigid plastic-laminated sheet metals. Anisotropic material properties were taken into account in [11]. Deformation theories of plasticity were employed to analyze plane strain bending at large strains in [12,13]. The work in [14] dealt with finite plastic bending of a compressible elastoplastic strip with combined isotropic and kinematic hardening at a given stretch normal to the bending plate. The flow theory of plasticity was adopted. Since the elastic portion of the strain tensor was not neglected in [14], the solution was much more complicated than those in [8,9,10,11]. It was shown in [15] that the assumption of incompressibility retained the simplicity of the rigid plastic solutions above for the corresponding elastic/plastic solutions. In particular, the Kirchhoff stress, which is involved in the equation that connects the elastic stretching and an objective stress rate, is equal to the Cauchy stress, and all objective stress rates reduce to the convected derivative. The authors of [15] adopted an inverse method that assumed transformation equations that connect Eulerian and Lagrangian coordinates, and showed that these equations described the geometry of plane strain bending at large strains. These equations can be used in conjunction with many constitutive equations [16,17]. The basic geometric assumptions used in [15] were similar to or even less restrictive than those accepted in papers published in journals devoted to metal forming [18,19].

The present paper extends the solution given in [17] to finite plane strain bending under tension of isotropic and kinematic hardening sheets. Prager’s law [20] is used to describe kinematic hardening. Isotropic hardening is controlled by an arbitrary function of the equivalent plastic strain. Elastic unloading is also included in the solution.

## 2. Constitutive Equations

A wide elastic/plastic sheet is subject to bending under tension by the bending moment *M* and the tensile force *F* under plane strain conditions. The initial cross-section of the sheet is rectangular (*A*_1_*B*_1_*C*_1_*D*_1_ in Figure 1a). The initial thickness and width of the sheet are *H* and 2*L*, respectively. The Eulerian Cartesian coordinate system (x, y) is chosen such that its *x*-axis coincides with the axis of symmetry of the cross-section throughout the process of deformation. The origin of the coordinate system is situated at point *O*, which is the intersection of the axis of symmetry and surface *AB* at any deformation stage (Figure 1b). The surface *AB* coincides with the plane *A*_1_*B*_1_ at the initial instant.

The commonly used Eulerian rate-type theory of elastoplasticity is adopted. This theory is presented, for example, in [21]. The Cauchy stress and Hencky strain are adopted throughout the paper. The material is assumed to be incompressible. Therefore, Poisson’s ratio is 1/2. Let $\tau_{1}$ and $\tau_{2}$ be the principal deviatoric stresses in the planes of flow. It will be seen later that all objective rates reduce to the convected derivative. Then, the elastic portions of the principal strain rates, $\xi_{1}^{e}$ and $\xi_{2}^{e}$, are given by:(1)$$\xi_{1}^{e}=\frac{{\dot{\tau }}_{1}}{2G}\;\mathrm{and}\;\xi_{2}^{e}=\frac{{\dot{\tau }}_{2}}{2G}$$
where the superimposed dot denotes the convected derivative and *G* is the shear modulus of elasticity. The plane strain criterion is assumed to be:(2)$$\left|\left(\tau_{1}-\alpha_{1}\right)-\left(\tau_{2}-\alpha_{2}\right)\right|=\frac{2\sigma_{0}}{\sqrt{3}}\Phi \left(\varepsilon_{eq}^{p}\right)$$
where $\alpha_{1}$ and $\alpha_{2}$ are the principal back stresses, $\varepsilon_{eq}^{p}$ is the equivalent plastic strain, $\sigma_{0}$ is the yield stress in tension at $\varepsilon_{eq}^{p}=0$, $\Phi \left(\varepsilon_{eq}^{p}\right)$ is an arbitrary function of its argument satisfying the conditions Φ(0)=1 and $d\Phi \left(\varepsilon_{eq}^{p}\right)/d\varepsilon_{eq}^{p}\geq 0$ for all $\varepsilon_{eq}^{p}$. The following equation defines the equivalent strain:(3)$${\dot{\varepsilon }}_{eq}^{p}=\sqrt{\frac{2}{3}}\sqrt{{\left(\xi_{1}^{p}\right)}^{2}+{\left(\xi_{2}^{p}\right)}^{2}}$$
where $\xi_{1}^{p}$ and $\xi_{2}^{p}$ are the plastic portions of the principal strain rates. The overdot in (3) denotes the convected derivative. In what follows, Prager’s law for the back stresses is accepted [20]. Then:(4)$${\dot{\alpha }}_{1}=K\xi_{1}^{p}\;\mathrm{and}\;{\dot{\alpha }}_{2}=K\xi_{2}^{p}$$
where *K* is a constitutive parameter. The plastic flow rule associated with the yield criterion (2) leads to:(5)$$\xi_{1}^{p}+\xi_{2}^{p}=0$$

The total strain rate tensor is the sum of the elastic and plastic strain rate tensors. Then:(6)$$\xi_{1}=\xi_{1}^{e}+\xi_{1}^{p}\;\mathrm{and}\;\xi_{2}=\xi_{2}^{e}+\xi_{2}^{p}$$

Equations (1), (5) and (6) combine to give the equation of incompressibility:(7)$$\xi_{1}^{}+\xi_{2}^{}=0$$

## 3. Strain Analysis

For a class of constitutive equations, strain analysis can be carried out independently of the stress equations. The compatibility of the strain distribution found with the stress equations should be verified *a posteriori*. This approach for pure plane strain bending has been proposed in [15] and extended to plane strain bending under tension in [17]. These papers have not dealt with kinematic hardening. However, one can try to combine the strain analysis completed there and kinematic hardening laws. For the reason of readability, the main results of [15,17] are presented in this section. These results are also necessary for finding the stress solution in the next section.

In addition to the Cartesian coordinate system (x, y), one can introduce the Lagrangian coordinate system (ζ, η) such that:(8)x=Hζ and y=Hη
at the initial instant. Then, it is seen from Figure 1 that ζ=0 on *AB*, ζ=−1 on *CD*, η=−L/H on *AD*, and η=L/H on *CB* throughout the process of deformation. The mapping between the (x, y) and (ζ, η) coordinate systems proposed in [15] is:(9)$$\frac{x}{H}=\sqrt{\frac{\zeta }{a}+\frac{s}{a^{2}}}\cos\left(2a\eta \right)-\frac{\sqrt{s}}{a},\;\frac{y}{H}=\sqrt{\frac{\zeta }{a}+\frac{s}{a^{2}}}\sin\left(2a\eta \right)$$
where *a* is a time-like variable such that a=0 at the initial instant and *s* is a function of *a*. This function should be found from the stress solution. The condition (8) is satisfied if:(10)$$s=\frac{1}{4}$$
at a=0. It is possible to verify by inspection that the mapping given by (9) and (10) satisfy Equation (7). Moreover, *AB* and *CD* are circular arcs, *CB* and *AD* are straight, and the coordinate curves of the (ζ, η)− coordinate system are principal strain rate and principal strain trajectories. Then, as noted in the previous section, all objective rates reduce to the convected derivative. The normal strains referred to (ζ, η)− coordinate system are determined from (9) as:(11)$$\varepsilon_{\zeta }=-\varepsilon_{\eta }=-\frac{1}{2}\ln\left[4\left(\zeta a+s\right)\right]$$

These strains are the principal strains. The equations in (4), (5) and (6) can be immediately integrated to get:(12)$$\alpha_{\zeta }=K\varepsilon_{\zeta }^{p}\;\mathrm{and}\;\alpha_{\eta }=K\varepsilon_{\eta }^{p}$$
(13)$$\varepsilon_{\zeta }^{p}=-\varepsilon_{\eta }^{p}$$
and:(14)$$\varepsilon_{\zeta }=\varepsilon_{\zeta }^{e}+\varepsilon_{\zeta }^{p}\;\mathrm{and}\;\varepsilon_{\eta }=\varepsilon_{\eta }^{e}+\varepsilon_{\eta }^{p}$$

Equations (3) and (5) combine to give ${\dot{\varepsilon }}_{eq}^{p}=2\left|\xi_{\zeta }^{p}\right|/\sqrt{3}=2\left|\xi_{\eta }^{p}\right|/\sqrt{3}$. As before, one can integrate these equations to get:(15)$$\begin{array}{c} \varepsilon_{eq}^{p}=\frac{2\varepsilon_{\zeta }^{p}}{\sqrt{3}}\;\mathrm{if}\;\xi_{\zeta }^{p}\geq 0\\ \varepsilon_{eq}^{p}=\frac{2\varepsilon_{\eta }^{p}}{\sqrt{3}}\;\mathrm{if}\;\xi_{\eta }^{p}\geq 0 \end{array}$$The inequalities should be satisfied throughout the process of deformation. Using (9), one can readily find that:(16)$$\frac{r_{AB}}{H}=\frac{\sqrt{s}}{a},\;\frac{r_{CD}}{H}=\frac{\sqrt{s-a}}{a},\;\theta_{0}=\frac{2aL}{H},\;\mathrm{and}\;\frac{h}{H}=\frac{\sqrt{s}-\sqrt{s-a}}{a}$$
where $r_{AB}$ is the radius of arc *AB*, $r_{CD}$ is the radius of arc *CD*, $\theta_{0}$ is the orientation of line *CB* relative to the *x*-axis, and *h* is the current thickness of the sheet (Figure 1).

The methodology of combining the strain analysis above and constitutive equations is provided in Appendix A.

## 4. Stress Analysis

The constitutive equations formulated in Section 2 result in the coaxiality of stress and strain rate tensors. Therefore, the coordinate curves of the Lagrangian coordinate system coincide with principal stress trajectories. In particular, the surfaces *AB*, *CD*, *CB* and *AD* are free of shear stresses. Moreover, the normal stresses referred to the (ζ, η)− coordinate system, $\sigma_{\zeta }$ and $\sigma_{\eta }$, are the principal stresses. Using (12) and (13), one can rewrite the yield criterion in Equation (2) as:(17)$$\left|\sigma_{\zeta }-\sigma_{\eta }-2K\varepsilon_{\zeta }^{p}\right|=\left|\sigma_{\zeta }-\sigma_{\eta }+2K\varepsilon_{\eta }^{p}\right|=\frac{2\sigma_{0}}{\sqrt{3}}\Phi \left(\varepsilon_{eq}^{p}\right)$$

It is assumed that the principal stresses are independent of η. Then, the only stress equilibrium equation that is not identically satisfied in the Lagrangian coordinates reads:(18)$$\frac{\partial \sigma_{\zeta }}{\partial \zeta }+\frac{a\left(\sigma_{\zeta }-\sigma_{\eta }\right)}{2\left(\zeta a+s\right)}=0$$

Equilibrium demands that some pressure *P* applies over the surface *CD* if F≠0 (Figure 1b). It is assumed that this pressure is uniformly distributed. Then, $F=r_{CD}P$. Using (16), one determines:(19)$$p=\frac{fa}{\sqrt{s-a}}$$
where $p=P/\sigma_{0}$ and $f=F/\left(\sigma_{0}H\right)$.

The stress boundary conditions are:(20)$$\sigma_{\zeta }=0$$
for ζ=0 and $\sigma_{\zeta }=-P$ for ζ=−1. Using (19), one transforms the latter to:(21)$$\frac{\sigma_{\zeta }}{\sigma_{0}}=-\frac{fa}{\sqrt{s-a}}$$
for ζ=−1.

The dimensionless bending moment, *m*, is determined as [17]:(22)$$m=\frac{2\sqrt{3}M}{\sigma_{0}H^{2}}=\frac{\sqrt{3}}{a}\int_{-1}^{0}\left(\frac{\sigma_{\eta }}{\sigma_{0}}-f\frac{H}{h}\right)d\zeta$$
where H/h should be eliminated by means of (16).

### 4.1. Purely Elastic Solution (Stage 1 of the Process)

The purely elastic solution is independent of the yield criterion. Therefore, the solution given in [17] is valid. In particular, the general solution in any elastic region is:(23)$$\frac{\sigma_{\zeta }}{\sigma_{0}}=\frac{1}{2\alpha }\ln^{2}\left[4\left(\zeta a+s\right)\right]+D_{0},\;\frac{\sigma_{\eta }}{\sigma_{0}}=\frac{1}{2\alpha }\ln^{2}\left[4\left(\zeta a+s\right)\right]+\frac{2}{\alpha }\ln\left[4\left(\zeta a+s\right)\right]+D_{0}$$
where $\alpha =\sigma_{0}/G$ and $D_{0}$ is constant. In deriving this result, it has been taken into account that $\tau_{1}=\tau_{\zeta }$, $\tau_{2}=\tau_{\eta }$, $\tau_{\zeta }=\sigma_{\zeta }-\sigma$, $\tau_{\eta }=\sigma_{\eta }-\sigma$, and $\sigma =\left(\sigma_{\zeta }+\sigma_{\eta }\right)/2$. In particular, Equation (1) after integrating becomes:(24)$$\varepsilon_{\zeta }^{e}=\frac{\left(\sigma_{\zeta }-\sigma_{\eta }\right)}{4G}\;\mathrm{and}\;\varepsilon_{\eta }^{e}=\frac{\left(\sigma_{\eta }-\sigma_{\zeta }\right)}{4G}$$

The purely elastic solution should satisfy the boundary conditions (20) and (21). As a result, it follows from (23) that:(25)$$D_{0}=-\frac{1}{2\alpha }\ln^{2}\left(4s\right)\;\mathrm{and}\;\frac{2\alpha fa}{\sqrt{s-a}}=\ln^{2}\left(4s\right)-\ln^{2}\left[4\left(s-a\right)\right]$$

At given values of a and *f*, the solution of the second equation in (25) supplies *s* as a function of *a*. Since $\varepsilon_{\zeta }^{p}=\varepsilon_{\eta }^{p}=\varepsilon_{eq}^{p}=0$ during this stage of the process, Equations (17), (23) and (25) show that plastic yielding initiates at ζ=0. The corresponding values of *a* and *s* are denoted as $a_{e}$ and $s_{e}$, respectively. One can find from (17) and (23) that $4s_{e}=\exp\left(\alpha /\sqrt{3}\right)$. Replacing *s* with $s_{e}$ in the second equation in (25) and solving the resulting equation for *a* provides $a_{e}$. In what follows, it is assumed that $a\geq a_{e}$.

### 4.2. Elastic/Plastic Solution with One Plastic Region (Stage 2 of the Process)

There are one elastic region, $-1\leq \zeta \leq \zeta_{1}$, and one plastic region, $\zeta_{1}\leq \zeta \leq 0$, during this stage of the process. Thus $\zeta_{1}$ is the elastic/plastic boundary. It is evident that $\zeta_{1}$ is a function of *a* and $\zeta_{1}=0$ at $a=a_{e}$. The solution (23) is valid in the elastic region, but (25) is not. The boundary condition (21) and the solution (23) combine to give:(26)$$D_{0}=-\frac{fa}{\sqrt{s-a}}-\frac{1}{2\alpha }\ln^{2}\left[4\left(s-a\right)\right]$$

It follows from (23) and (26) that:(27)$$\begin{array}{l} \frac{\sigma_{\zeta }}{\sigma_{0}}=\frac{p_{2}}{\sigma_{0}}=\frac{1}{2\alpha }\ln^{2}\left[4\left(\zeta_{1}a+s\right)\right]-\frac{fa}{\sqrt{s-a}}-\frac{1}{2\alpha }\ln^{2}\left[4\left(s-a\right)\right],\;\\ \frac{\sigma_{\eta }-\sigma_{\zeta }}{\sigma_{0}}=\frac{2}{\alpha }\ln\left[4\left(\zeta_{1}a+s\right)\right] \end{array}$$
at $\zeta =\zeta_{1}$.

The distribution of $\sigma_{\zeta }$ and $\sigma_{\eta }$ in the plastic region is found from (17) and (18). In this region, $\sigma_{\eta }>\sigma_{\zeta }$ and $\xi_{\eta }^{p}>0$. Therefore, Equation (15) becomes:(28)$$\varepsilon_{eq}^{p}=\frac{2\varepsilon_{\eta }^{p}}{\sqrt{3}}$$

Moreover, using (28), one transforms Equation (17) to:(29)$$\sigma_{\eta }-\sigma_{\zeta }-\sqrt{3}K\varepsilon_{eq}^{p}=\frac{2\sigma_{0}}{\sqrt{3}}\Phi \left(\varepsilon_{eq}^{p}\right)$$

Equations (18) and (29) combine to give:(30)$$\frac{\partial \sigma_{\zeta }}{\partial \zeta }=\frac{\sqrt{3}a}{2\left(\zeta a+s\right)}\left[\beta \varepsilon_{eq}^{p}+\frac{2}{3}\Phi \left(\varepsilon_{eq}^{p}\right)\right]$$
where $\beta =K/\sigma_{0}$. The elastic strains are determined from (11), (14) and (28) as:(31)$$\varepsilon_{\zeta }^{e}=-\frac{1}{2}\ln\left[4\left(\zeta a+s\right)\right]+\frac{\sqrt{3}}{2}\varepsilon_{eq}^{p}\;\mathrm{and}\;\varepsilon_{\eta }^{e}=\frac{1}{2}\ln\left[4\left(\zeta a+s\right)\right]-\frac{\sqrt{3}}{2}\varepsilon_{eq}^{p}$$

Equations (24), (29) and (31) combine to give:(32)$$\ln\left[4\left(\zeta a+s\right)\right]=\sqrt{3}\left(1+\frac{\alpha \beta }{2}\right)\varepsilon_{eq}^{p}+\frac{\alpha }{\sqrt{3}}\Phi \left(\varepsilon_{eq}^{p}\right)$$

It follows from this equation that
(33)$$\frac{\partial \varepsilon_{eq}^{p}}{\partial \zeta }=a{\left(\zeta a+s\right)}^{-1}{\left[\sqrt{3}\left(1+\frac{\alpha \beta }{2}\right)+\frac{\alpha }{\sqrt{3}}\Phi^{\prime }\left(\varepsilon_{eq}^{p}\right)\right]}^{-1}$$
where $\Phi^{\prime }\left(\varepsilon_{eq}^{p}\right)=d\Phi /d\varepsilon_{eq}^{p}$. One replaces differentiation with respect to ζ with differentiation with respect to $\varepsilon_{eq}^{p}$ in (30) using (33) to transform (30) to
(34)$$\frac{\partial \sigma_{\zeta }}{\partial \varepsilon_{eq}^{p}}=\frac{\sqrt{3}}{2}\left[\beta \varepsilon_{eq}^{p}+\frac{2}{3}\Phi \left(\varepsilon_{eq}^{p}\right)\right]\left[\sqrt{3}\left(1+\frac{\alpha \beta }{2}\right)+\frac{\alpha }{\sqrt{3}}\Phi^{\prime }\left(\varepsilon_{eq}^{p}\right)\right]$$

Let $\varepsilon_{1}$ be the value of $\varepsilon_{eq}^{p}$ at ζ=0. Then, the boundary condition (20) becomes:(35)$$\sigma_{\zeta }=0$$
for $\varepsilon_{eq}^{p}=\varepsilon_{1}$. Both $\sigma_{\zeta }$ and $\sigma_{\eta }$ should be continuous across the elastic/plastic boundary. Moreover, $\varepsilon_{eq}^{p}=0$ at this boundary. Therefore, the solution of Equation (34) satisfying the condition that $\sigma_{\zeta }$ is continuous is
(36)$$\frac{\sigma_{\zeta }}{\sigma_{0}}=\frac{3\beta }{4}\left(1+\frac{\alpha \beta }{2}\right){\left(\varepsilon_{eq}^{p}\right)}^{2}+\Psi \left(\varepsilon_{eq}^{p}\right)-\Psi \left(0\right)+\frac{\alpha }{6}\left[\Phi^{2}\left(\varepsilon_{eq}^{p}\right)-1\right]+\frac{\alpha \beta }{2}\varepsilon_{eq}^{p}\Phi \left(\varepsilon_{eq}^{p}\right)+\frac{p_{2}}{\sigma_{0}}$$
where $\Psi \left(\varepsilon_{eq}^{p}\right)$ is the antiderivative of $\Phi \left(\varepsilon_{eq}^{p}\right)$ and $p_{2}$ should be eliminated using (27). The stress $\sigma_{\eta }$ is determined from (29) and (36) as:(37)$$\begin{array}{l} \frac{\sigma_{\eta }}{\sigma_{0}}=\frac{3\beta }{4}\left(1+\frac{\alpha \beta }{2}\right){\left(\varepsilon_{eq}^{p}\right)}^{2}+\Psi \left(\varepsilon_{eq}^{p}\right)-\Psi \left(0\right)+\frac{\alpha }{6}\left[\Phi^{2}\left(\varepsilon_{eq}^{p}\right)-1\right]+\\ \;\frac{\alpha \beta }{2}\varepsilon_{eq}^{p}\Phi \left(\varepsilon_{eq}^{p}\right)+\sqrt{3}\beta \varepsilon_{eq}^{p}+\frac{2}{\sqrt{3}}\Phi \left(\varepsilon_{eq}^{p}\right)+\frac{p_{2}}{\sigma_{0}} \end{array}$$

The continuity of $\sigma_{\eta }$ across the elastic/plastic boundary is equivalent to the continuity of $\sigma_{\eta }-\sigma_{\zeta }$. The latter combined with (27) and (29) at $\varepsilon_{eq}^{p}=0$ leads to
(38)$$\zeta_{1}=\frac{1}{a}\left[\frac{1}{4}\exp\left(\frac{\alpha }{\sqrt{3}}\right)-s\right]$$

Using the boundary condition (35), one finds from (36) that
(39)$$\frac{3\beta }{4}\left(1+\frac{\alpha \beta }{2}\right)\varepsilon_{1}^{2}+\Psi \left(\varepsilon_{1}\right)-\Psi \left(0\right)+\frac{\alpha }{6}\left[\Phi^{2}\left(\varepsilon_{1}\right)-1\right]+\frac{\alpha \beta }{2}\varepsilon_{1}\Phi \left(\varepsilon_{1}\right)+\frac{p_{2}}{\sigma_{0}}=0$$

It follows from (32) that
(40)$$\ln\left(4s\right)=\sqrt{3}\left(1+\frac{\alpha \beta }{2}\right)\varepsilon_{1}+\frac{\alpha }{\sqrt{3}}\Phi \left(\varepsilon_{1}\right)$$

Using this equation, one eliminates *s* in the expression for $p_{2}$ given in (27). As a result, Equation (39) connects *a* and $\varepsilon_{1}$. This equation should be solved numerically to find the dependence of $\varepsilon_{1}$ on *a*. Then, *s*, $\zeta_{1}$, and $D_{0}$ as functions of *a* are readily determined from (40), (38) and (26). The equations in (23) supply the distribution of the stresses in the elastic region. Equations (32), (36) and (37) provide the distribution of the stresses in the plastic region in parametric form with $\varepsilon_{eq}^{p}$ being the parameter. The bending moment is determined from (22) by numerical integration.

This stage of the process ends when the yield criterion starts to satisfy at ζ=−1. It follows from (23) and (17) at $\varepsilon_{\zeta }^{p}=\varepsilon_{\eta }^{p}=0$ that:(41)$$\left|\ln\left[4\left(s-a\right)\right]\right|=\frac{\alpha }{\sqrt{3}}$$

Since *s* as a function of *a* has already been found, this equation allows one to determine the value of *a* corresponding to plastic yielding initiation at ζ=−1. This value is denoted as $a_{1}$.

### 4.3. Elastic/plastic Solution with Two Plastic Regions (Stage 3 of the Process)

There are one elastic region, $\zeta_{2}\leq \zeta \leq \zeta_{1}$, and two plastic regions, $\zeta_{1}\leq \zeta \leq 0$ and $-1\leq \zeta \leq \zeta_{2}$, at $a>a_{1}$. Thus $\zeta_{1}$ and $\zeta_{2}$ are the elastic/plastic boundaries. Both are functions of *a* and $\zeta_{2}=-1$ at $a=a_{1}$. The solution (23) is valid in the elastic region, but (25) and (26) are not. Moreover, the general solution in the plastic region $\zeta_{1}\leq \zeta \leq 0$ provided in Section 4.2 is also valid. 

Consider the plastic region $-1\leq \zeta \leq \zeta_{2}$. In this region, $\sigma_{\eta }<\sigma_{\zeta }$ and $\xi_{\zeta }^{p}>0$. Therefore, Equation (15) becomes:(42)$$\varepsilon_{eq}^{p}=\frac{2\varepsilon_{\zeta }^{p}}{\sqrt{3}}$$

Moreover, using (42), one transforms Equation (17) to:(43)$$\sigma_{\zeta }-\sigma_{\eta }-\sqrt{3}K\varepsilon_{eq}^{p}=\frac{2\sigma_{0}}{\sqrt{3}}\Phi \left(\varepsilon_{eq}^{p}\right)$$

Equations (18) and (43) combine to give:(44)$$\frac{\partial \sigma_{\zeta }}{\partial \zeta }=-\frac{\sqrt{3}a}{2\left(\zeta a+s\right)}\left[\beta \varepsilon_{eq}^{p}+\frac{2}{3}\Phi \left(\varepsilon_{eq}^{p}\right)\right]$$

The elastic strains are determined from (11), (14) and (42) as:(45)$$\varepsilon_{\zeta }^{e}=-\frac{1}{2}\ln\left[4\left(\zeta a+s\right)\right]-\frac{\sqrt{3}}{2}\varepsilon_{eq}^{p}\;\mathrm{and}\;\varepsilon_{\eta }^{e}=\frac{1}{2}\ln\left[4\left(\zeta a+s\right)\right]+\frac{\sqrt{3}}{2}\varepsilon_{eq}^{p}$$

Equations (24), (43) and (45) combine to give:(46)$$\ln\left[4\left(\zeta a+s\right)\right]=-\sqrt{3}\left(1+\frac{\alpha \beta }{2}\right)\varepsilon_{eq}^{p}-\frac{\alpha }{\sqrt{3}}\Phi \left(\varepsilon_{eq}^{p}\right)$$

It follows from this equation that:(47)$$\frac{\partial \varepsilon_{eq}^{p}}{\partial \zeta }=-a{\left(\zeta a+s\right)}^{-1}{\left[\sqrt{3}\left(1+\frac{\alpha \beta }{2}\right)+\frac{\alpha }{\sqrt{3}}\Phi^{\prime }\left(\varepsilon_{eq}^{p}\right)\right]}^{-1}$$

One replaces differentiation with respect to ζ with differentiation with respect to $\varepsilon_{eq}^{p}$ in (44) using (47) to transform (44) to:(48)$$\frac{\partial \sigma_{\zeta }}{\partial \varepsilon_{eq}^{p}}=\frac{\sqrt{3}}{2}\left[\beta \varepsilon_{eq}^{p}+\frac{2}{3}\Phi \left(\varepsilon_{eq}^{p}\right)\right]\left[\sqrt{3}\left(1+\frac{\alpha \beta }{2}\right)+\frac{\alpha }{\sqrt{3}}\Phi^{\prime }\left(\varepsilon_{eq}^{p}\right)\right]$$

Let $\varepsilon_{2}$ be the value of $\varepsilon_{eq}^{p}$ at ζ=−1. Then, the boundary condition (21) becomes:(49)$$\frac{\sigma_{\zeta }}{\sigma_{0}}=-\frac{fa}{\sqrt{s-a}}$$
for $\varepsilon_{eq}^{p}=\varepsilon_{2}$. Both $\sigma_{\zeta }$ and $\sigma_{\eta }$ should be continuous across the elastic/plastic boundary $\zeta =\zeta_{2}$. Moreover, $\varepsilon_{eq}^{p}=0$ at this boundary. Therefore, the solution of Equation (48) satisfying the condition that $\sigma_{\zeta }$ is continuous is:(50)$$\frac{\sigma_{\zeta }}{\sigma_{0}}=\frac{3\beta }{4}\left(1+\frac{\alpha \beta }{2}\right){\left(\varepsilon_{eq}^{p}\right)}^{2}+\Psi \left(\varepsilon_{eq}^{p}\right)-\Psi \left(0\right)+\frac{\alpha }{6}\left[\Phi^{2}\left(\varepsilon_{eq}^{p}\right)-1\right]+\frac{\alpha \beta }{2}\varepsilon_{eq}^{p}\Phi \left(\varepsilon_{eq}^{p}\right)+\frac{p_{1}}{\sigma_{0}}$$
where $p_{1}$ is determined from (23) as:(51)$$\frac{p_{1}}{\sigma_{0}}=\frac{1}{2\alpha }\ln^{2}\left[4\left(\zeta_{2}a+s\right)\right]+D_{0}$$

The stress $\sigma_{\eta }$ is found from (43) and (50) as:(52)$$\begin{array}{l} \frac{\sigma_{\eta }}{\sigma_{0}}=\frac{3\beta }{4}\left(1+\frac{\alpha \beta }{2}\right){\left(\varepsilon_{eq}^{p}\right)}^{2}+\Psi \left(\varepsilon_{eq}^{p}\right)-\Psi \left(0\right)+\frac{\alpha }{6}\left[\Phi^{2}\left(\varepsilon_{eq}^{p}\right)-1\right]\\ \;+\frac{\alpha \beta }{2}\varepsilon_{eq}^{p}\Phi \left(\varepsilon_{eq}^{p}\right)-\sqrt{3}\beta \varepsilon_{eq}^{p}-\frac{2}{\sqrt{3}}\Phi \left(\varepsilon_{eq}^{p}\right)+\frac{p_{1}}{\sigma_{0}} \end{array}$$

The continuity of $\sigma_{\eta }$ across the elastic/plastic boundary is equivalent to the continuity of $\sigma_{\eta }-\sigma_{\zeta }$. The latter combined with (23) and (43) at $\varepsilon_{eq}^{p}=0$ leads to:(53)$$\zeta_{2}=\frac{1}{a}\left[\frac{1}{4}\exp\left(-\frac{\alpha }{\sqrt{3}}\right)-s\right]$$

Using the boundary condition (49), one finds from (50) that:(54)$$\frac{3\beta }{4}\left(1+\frac{\alpha \beta }{2}\right){\left(\varepsilon_{2}\right)}^{2}+\Psi \left(\varepsilon_{2}\right)-\Psi \left(0\right)+\frac{\alpha }{6}\left[\Phi^{2}\left(\varepsilon_{2}\right)-1\right]+\frac{\alpha \beta }{2}\varepsilon_{2}\Phi \left(\varepsilon_{2}\right)+\frac{p_{1}}{\sigma_{0}}=-\frac{fa}{\sqrt{s-a}}$$

It follows from (46) that:(55)$$\ln\left[4\left(s-a\right)\right]=-\sqrt{3}\left(1+\frac{\alpha \beta }{2}\right)\varepsilon_{2}-\frac{\alpha }{\sqrt{3}}\Phi \left(\varepsilon_{2}\right)$$

The first equation in (23) supplies the value of $p_{2}$ involved in (36) and (37) in the form:(56)$$\frac{p_{2}}{\sigma_{0}}=\frac{\alpha }{6}+D_{0}$$

Here Equation (38) has been taken into account. Equations (51) and (53) combine to give:(57)$$\frac{p_{1}}{\sigma_{0}}=\frac{\alpha }{6}+D_{0}$$

It is seen from (56) and (57) that $p_{1}=p_{2}$ during this stage of the process. Therefore, one can replace $p_{2}$ with $p_{1}$ in (36), (37) and (39). Then, one eliminates $p_{1}$ between (39) and (54) to get:(58)$$\begin{array}{l} \frac{3\beta }{4}\left(1+\frac{\alpha \beta }{2}\right)\varepsilon_{1}^{2}+\Psi \left(\varepsilon_{1}\right)-\Psi \left(0\right)+\frac{\alpha }{6}\left[\Phi^{2}\left(\varepsilon_{1}\right)-1\right]+\frac{\alpha \beta }{2}\varepsilon_{1}\Phi \left(\varepsilon_{1}\right)=\\ \frac{3\beta }{4}\left(1+\frac{\alpha \beta }{2}\right){\left(\varepsilon_{2}\right)}^{2}+\Psi \left(\varepsilon_{2}\right)-\Psi \left(0\right)+\frac{\alpha }{6}\left[\Phi^{2}\left(\varepsilon_{2}\right)-1\right]+\frac{\alpha \beta }{2}\varepsilon_{2}\Phi \left(\varepsilon_{2}\right)+\frac{fa}{\sqrt{s-a}} \end{array}$$

Equations (40) and (55) are solved for *a* and *s* as:(59)$$\begin{array}{l} s=\frac{1}{4}\exp\left[\sqrt{3}\left(1+\frac{\alpha \beta }{2}\right)\varepsilon_{1}+\frac{\alpha }{\sqrt{3}}\Phi \left(\varepsilon_{1}\right)\right],\\ a=\frac{1}{4}\exp\left[\sqrt{3}\left(1+\frac{\alpha \beta }{2}\right)\varepsilon_{1}+\frac{\alpha }{\sqrt{3}}\Phi \left(\varepsilon_{1}\right)\right]-\frac{1}{4}\exp\left[-\sqrt{3}\left(1+\frac{\alpha \beta }{2}\right)\varepsilon_{2}-\frac{\alpha }{\sqrt{3}}\Phi \left(\varepsilon_{2}\right)\right] \end{array}$$

Using (59), one eliminates *a* and *s* in (58). The resulting equation connects $\varepsilon_{2}$ and $\varepsilon_{1}$. This equation should be solved numerically to find the dependence of $\varepsilon_{1}$ on $\varepsilon_{2}$. Then, *a*, *s*, $\zeta_{1}$, $\zeta_{2}$, $p_{1}$, and $D_{0}$ as functions of $\varepsilon_{2}$ are readily determined from (59), (38), (53), (54), and (57), respectively. It is evident that it is also possible to express $\varepsilon_{1}$, $\varepsilon_{2}$, *s*, $\zeta_{1}$, $\zeta_{2}$, $p_{1}$, and $D_{0}$ as functions of *a* using this solution. The equations in (23) supply the distribution of the stresses in the elastic region. Equations (32), (36) and (37) provide the distribution of the stresses in the plastic region $\zeta_{1}\leq \zeta \leq 0$ in parametric form with $\varepsilon_{eq}^{p}$ being the parameter. Equations (46), (50) and (52) provide the distribution of the stresses in the plastic region $-1\leq \zeta \leq \zeta_{2}$ in parametric form with $\varepsilon_{eq}^{p}$ being the parameter. The bending moment is determined from (22) by numerical integration.

The solution above is valid if the size of each plastic region increases as the deformation proceeds. The condition is equivalent to:(60)$$\frac{d\zeta_{1}}{da}\leq 0\;\mathrm{and}\;\frac{d\zeta_{2}}{da}\geq 0$$

These inequalities should be verified numerically. It is worthy to note that the process’ analysis beyond the instant when one of the conditions in (60) is violated is possible but requires additional constitutive equations that describe reversed plastic yielding.

## 5. Residual Stresses

Consider elastic unloading, assuming that the strains and the displacements are small at this stage. Therefore, variations of the shape can be neglected. Denote $r_{CD}=R_{0}$ and $r_{AB}=R_{1}$ at the end of loading when $a=a_{l}$, $s=s_{l}$, $p=p_{l}$, $h=h_{l}$ and $m=m_{l}$. All these values are calculated using the solution given in the previous section. The distribution of $\sigma_{\zeta }$ and $\sigma_{\eta }$ corresponding to $a=a_{l}$ is denoted as $\sigma_{\zeta }^{\left(l\right)}$ and $\sigma_{\eta }^{\left(l\right)}$. It is convenient to introduce a polar coordinate system (r, θ) by the following transformation equations:(61)$$\frac{r}{H}=\frac{\sqrt{\zeta a_{l}+s_{l}}}{a_{l}},\;\theta =2a_{l}\eta$$

It is evident from these equations that the coordinate curves of the (r, θ)—coordinate system coincide with the coordinate curves of the (ζ, η)—coordinate system. Therefore, $\sigma_{\zeta }^{\left(l\right)}=\sigma_{r}$ and $\sigma_{\eta }^{\left(l\right)}=\sigma_{\theta }$. Using (61), one can represent the solution given in the previous section in the polar coordinate system with no difficulty.

The general solution for the increment of the principal stresses from the configuration corresponding to $a=a_{l}$ is independent of the solution at loading. Therefore, one may adopt the solution provided in [17]. According to this solution:(62)$$\frac{\sigma_{r}^{*}}{2G}=V_{0}\ln\left(\frac{r}{R_{0}}\right)-U_{0}{\left(\frac{R_{0}}{r}\right)}^{2}+C_{1},\;\;\frac{\sigma_{\theta }^{*}}{2G}=V_{0}\left[1+\ln\left(\frac{r}{R_{0}}\right)\right]+U_{0}{\left(\frac{R_{0}}{r}\right)}^{2}+C_{1}$$
where
(63)$$\begin{array}{l} U_{0}=C_{1}-\frac{\alpha p_{l}}{2},\;V_{0}=\frac{2C_{1}\left(1-\rho_{0}^{2}\right)+\alpha \rho_{0}^{2}p_{l}}{2\ln\rho_{0}},\\ C_{1}=-B/A,\\ B=\frac{\alpha f\ln\rho_{0}}{h_{l}}-\frac{\alpha p_{l}\rho_{0}^{4}}{2\sigma_{0}}+\frac{\alpha \rho_{0}^{2}}{6}\left[3p_{l}+2\ln\rho_{0}\left(\sqrt{3}m_{l}\frac{H^{2}}{R_{0}^{2}}-3\frac{f}{h_{l}}-3p_{l}+6p_{l}\ln\rho_{0}\right)\right],\\ A={\left(\rho_{0}^{2}-1\right)}^{2}-4\rho_{0}^{2}\ln^{2}\rho_{0} \end{array}$$
where $\rho_{0}=R_{0}/R_{1}$. The radial distribution of the residual stresses can be found from the equations $\sigma_{r}^{res}=\sigma_{\zeta }^{res}=\sigma_{r}+\sigma_{r}^{*}$ and $\sigma_{\theta }^{res}=\sigma_{\theta }+\sigma_{\theta }^{*}$.

The solution in this section is valid if no reverse yielding occurs. Using Equation (17), the corresponding condition can be written in the form:(64)$$\Lambda =\left|\frac{\sigma_{\zeta }^{res}}{\sigma_{0}}-\frac{\sigma_{\eta }^{res}}{\sigma_{0}}-2\beta \varepsilon_{\zeta }^{p}\right|-\frac{2}{\sqrt{3}}\Phi \left(\varepsilon_{eq}^{p}\right)\leq 0$$

In this equation, $\varepsilon_{eq}^{p}$ and $\varepsilon_{\zeta }^{p}$ are understood to be calculated at $a=a_{l}$.

## 6. Numerical Example

Even though the solution found is semianalytic, its full parametric analysis is not feasible due to the significant number of parameters that affect the solution. Since the solution for isotropic hardening laws has already been discussed in [17], the numerical example below focuses on kinematic hardening. The numerical integration and solution of transcendental equations have been performed using the corresponding build-in commands in Mathematica (version 11.3) [22].

Swift’s law describes isotropic hardening. Then:(65)$$\Phi \left(\varepsilon_{eq}^{p}\right)={\left(1+\frac{\varepsilon_{eq}^{p}}{\varepsilon_{0}}\right)}^{n},\;\Psi \left(\varepsilon_{eq}^{p}\right)=\frac{\varepsilon_{0}}{\left(1+n\right)}{\left(1+\frac{\varepsilon_{eq}^{p}}{\varepsilon_{0}}\right)}^{n+1},\;\Phi^{\prime }\left(\varepsilon_{eq}^{p}\right)=\frac{n}{\varepsilon_{0}}{\left(1+\frac{\varepsilon_{eq}^{p}}{\varepsilon_{0}}\right)}^{n-1}$$

It is assumed that n=0.25 and $\varepsilon_{0}=0.222$. In all calculations, $\alpha =3\times 10^{-3}$. The parameter β varies in the range 0≤β≤10. The material with no kinematic hardening is obtained at β=0. The through-thickness distribution of the principal stresses and the bending moment are calculated according to the procedure described in Section 4.

The inequalities in (60) control the range of validity of the solution at loading. Figure 2 demonstrates the variation of $\zeta_{1}$ and $\zeta_{2}$ with $H/r_{CD}$ at β=5 for several values of *f*. The convex down lines correspond to $\zeta_{1}$ and the convex up lines to $\zeta_{2}$. Stars denote the extremum points. It is evident that the derivatives with respect to *a* and $H/r_{CD}$ have the same sign and vanish simultaneously. It is seen from Figure 2 that the solution may break down because (i) $d\zeta_{1}/da=0$ and $d\zeta_{2}/da>0$, or (ii) $d\zeta_{2}/da=0$ and $d\zeta_{1}/da<0$, or (iii) $d\zeta_{1}/da=0$ and $d\zeta_{2}/da=0$ at a certain value of $H/r_{CD}$. These conditions are taken into account in the numerical example below.

Since $\zeta =\zeta_{1}$ and $\zeta =\zeta_{2}$ are the elastic/plastic boundaries, Figure 2 illustrates the elastic and plastic regions’ thickness in the Lagrangian coordinates. In particular, the thickness of the plastic region adjacent to the outside surface is $\left|\zeta_{1}\right|$, the thickness of the plastic region adjacent to the inside surface is $\left(1+\zeta_{2}\right)$, and the thickness of the elastic region is $\left(\zeta_{1}-\zeta_{2}\right)$. It is seen from this figure that the thickness of the elastic region decreases quickly at the very beginning of the process and, as a result, becomes small after a small amount of deformation.

Figure 3 and Figure 4 depict the variation of the dimensionless bending moment with $H/r_{CD}$ at f=0 (pure bending) and f=0.2, respectively. Each figure shows several curves corresponding to different values of β. It is evident that m=0 and $H/r_{CD}=0$ at the initial instant. Figure 3 and Figure 4 reveal a significant effect of β on the dimensionless bending moment. In general, the material with no kinematic hardening requires the least moment for the deformation to proceed.

Figure 5, Figure 6, Figure 7 and Figure 8 illustrate the through-thickness distribution of the principal stresses at *H*/*r_CD_* = 0.06 and several values of β. This value of corresponds *r_CD_* to the third stage of the process. In these figures, *X* is the dimensionless distance from the surface ζ = −1 defined as:(66)$$X=\frac{r-r_{CD}}{H}$$

The distribution of the stress $\sigma_{\zeta }$ at f=0 and f=0.2 is depicted in Figure 5 and Figure 7, respectively. Figure 6 and Figure 8 show the distribution of the stress $\sigma_{\eta }$ at the same set of parameters. It is seen from these figures that the thickness of the elastic region is quite narrow at this stage of the process. The effect of β on the magnitude of the stress $\sigma_{\eta }$ is not significant in some vicinity of the elastic/plastic boundaries and increases as the distance from these boundaries increases. The effect is most pronounced and significant at the surfaces ζ=−1 and ζ=0 and in the elastic region’s central part. The effect of β on the magnitude of the stress $\sigma_{\zeta }$ is most significant in the sheet’s central part. It is not surprising since the magnitude of this stress is prescribed at ζ=−1 and ζ=0 by the boundary conditions.

Figure 9, Figure 10, Figure 11 and Figure 12 illustrate the through-thickness distribution of the residual stresses calculated according to the procedure in Section 5. All the parameters corresponding to the end of loading and involved in this procedure are taken at $H/r_{CD}=0.06$. Therefore, the distributions of $\sigma_{\zeta }$ and $\sigma_{\eta }$ shown in Figure 5, Figure 6, Figure 7 and Figure 8 are $\sigma_{\zeta }^{\left(l\right)}$ and $\sigma_{\eta }^{\left(l\right)}$.

The variation of the stress $\sigma_{\zeta }^{res}$ with *X* at f=0 and f=0.2 is depicted in Figure 9 and Figure 11, respectively. Figure 10 and Figure 12 show the through-thickness distribution of the stress $\sigma_{\eta }^{res}$. The magnitude of the stress $\sigma_{\zeta }^{res}$ is quite small. The magnitude of the stress $\sigma_{\eta }^{res}$ sharply increases as *X* increases in the region that is purely elastic at the loading stage. This stress significantly decreases as *X* increases in the two other regions and vanishes at two points. One of these points lies in the region that is purely elastic at the loading stage, and the other in the region $\zeta_{1}<\zeta <0$. The latter region is plastic at the loading stage.

The distribution of the residual stresses does not satisfy the inequality (64) if β is large enough. It is evident that one can represent Λ involved in (64) as a function of *X*. This function is depicted in Figure 13 for f=0 and Figure 14 for f=0.2. It is seen from Figure 14 that Λ practically vanishes at one point if β=6. Therefore, the solution for the residual stresses is not valid for β>6. For this reason, the curves for β=8 and β=10 are not shown in Figure 11 and Figure 12. For consistency, such curves are not depicted in Figure 9 and Figure 10.

## 7. Discussion

A semianalytic solution describing the process of finite plane strain bending under tension of an elastic/plastic sheet has been derived assuming combined isotropic and kinematic hardening. The material is supposed to be incompressible. The solution has been found by an inverse method starting from the transformation equations between Eulerian and Lagrangian coordinates given in Equation (9). In general, the process consists of three stages. The sheet is purely elastic during the first stage. This stage ends when a plastic region starts to spread from the surface *AB* to the surface *CD* (Figure 1). The second stage of the process ends when another plastic region starts to spread from the surface *CD* to the surface *AB* (Figure 1). The solution is valid if the inequalities in Equation (60) are satisfied. Figure 2 shows that these solution nonexistence conditions depend on process parameters. The process of unloading has also been considered assuming that no reverse plastic region appears. Equation (64) controls the latter condition.

The solution is semianalytic but includes several parameters and an arbitrary function $\Phi \left(\varepsilon_{eq}^{p}\right),$ its derivative and antiderivative. Therefore, the complete parametric analysis of the solution is not feasible, but it is straightforward to find any quantity of interest for any given set of parameters and the function $\Phi \left(\varepsilon_{eq}^{p}\right)$. In particular, the derivative and antiderivative of $\Phi \left(\varepsilon_{eq}^{p}\right)$ can be found in terms of elementary functions for all widely used isotropic hardening laws.

One of the limiting cases of the process considered in the present paper is tension with no bending. In this case, there is one elastic region at the beginning of the process. This region is replaced with a plastic region at a certain value of the tensile force. The distribution of stresses and strains is uniform in this limiting case. If the bending moment is small, then Stage 3 of the process never occurs. The solutions given in Section 4.1 and Section 4.2 are valid. As the deformation proceeds, the elastic/plastic boundary may reach the inside surface of the sheet, $\zeta_{1}=-1$. If the deformation further proceeds, the plastic region occupies the entire sheet.

The numerical example has been provided for the function $\Phi \left(\varepsilon_{eq}^{p}\right)$ shown in Equation (65). Figure 3, Figure 4, Figure 5, Figure 6, Figure 7, Figure 8, Figure 9, Figure 10, Figure 11, Figure 12, Figure 13 and Figure 14 demonstrate that the parameter β involved in Prager’s law significantly affects the bending moment and the through-thickness distribution of the stresses at loading but slightly affects the through-thickness distribution of the residual stresses. Of course, this conclusion is only valid for the parameters used in the numerical solution.

The solution found in conjunction with experimental research can be used for identifying constitutive models. Various bending tests have already been employed for this purpose [2,3,4,5,6,7]. An advantage of the present solution is that it is given in closed form, except simple numerical techniques for calculating the bending moment and solving transcendental equations, for arbitrary function $\Phi \left(\varepsilon_{eq}^{p}\right)$ involved in Equation (2). One can vary this function for improving the agreement between the theory and experiment.

The solution can be used as a benchmark problem for verifying numerical codes, which a necessary step before using such codes [23,24].

## 8. Conclusions

The following conclusions were reached from the theoretical analysis of plane strain bending under tension of a wide sheet:(1)The constitutive equations comprising combined isotropic and kinematic hardening laws and the normality rule permit a semianalytic solution. No restriction, other than the conventional restrictions from the general theory of plasticity, is imposed on the isotropic hardening law. Prager’s law describes kinematic hardening. Numerical techniques are only necessary for solving transcendental equations and evaluating ordinary integrals.(2)An inverse method is used for finding the solution. Its key point is transformation equations that connect Eulerian and Lagrangian coordinates. These equations describe the geometry of the process and are independent of the stress equations.(3)The distribution of residual stresses is also given, assuming that unloading is purely elastic.(4)An applied aspect of the solution is that it can be used in conjunction with experimental data for identifying material properties. An advantage of the solution in this respect is that it is valid for the arbitrary isotropic hardening law.(5)One can use the solution for verifying numerical codes, which is a necessary step for their subsequent use.

## Figures and Tables

**Figure 1 materials-14-01166-f001:**
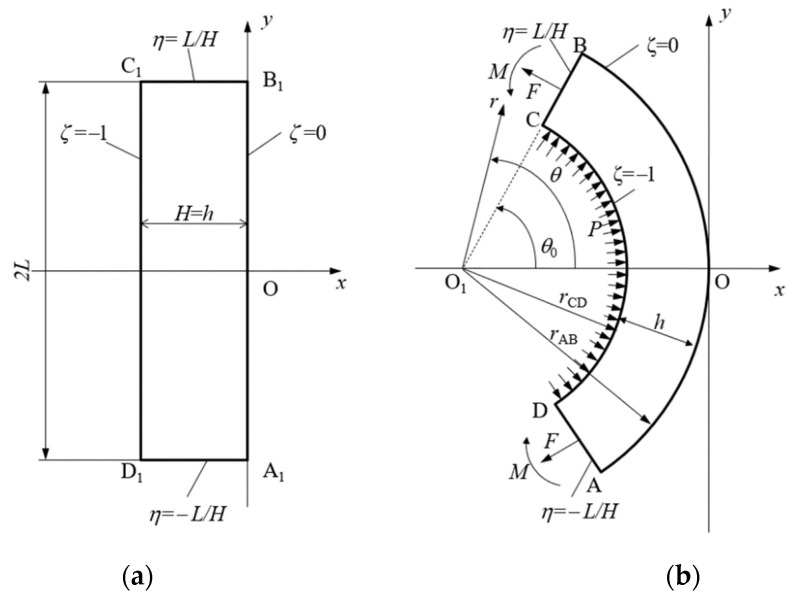
Schematic diagram of the process: (**a**) initial configuration, (**b**) intermediate and final configurations.

**Figure 2 materials-14-01166-f002:**
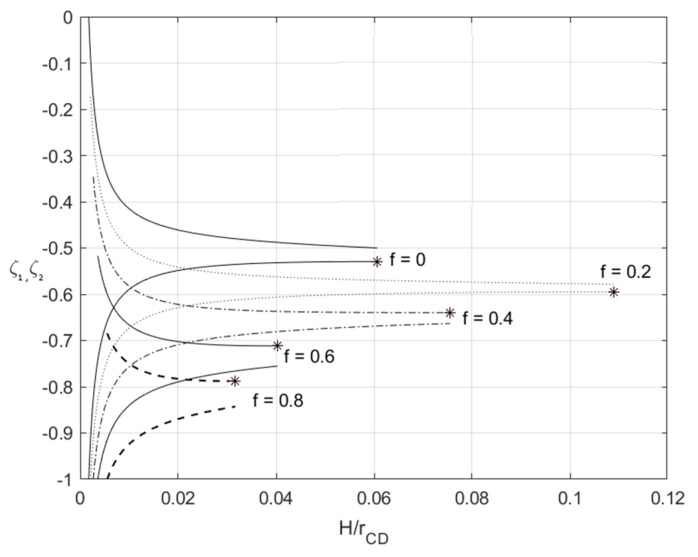
Illustration of the solution nonexistence conditions following from Equation (60) at β=5 and several values of f.

**Figure 3 materials-14-01166-f003:**
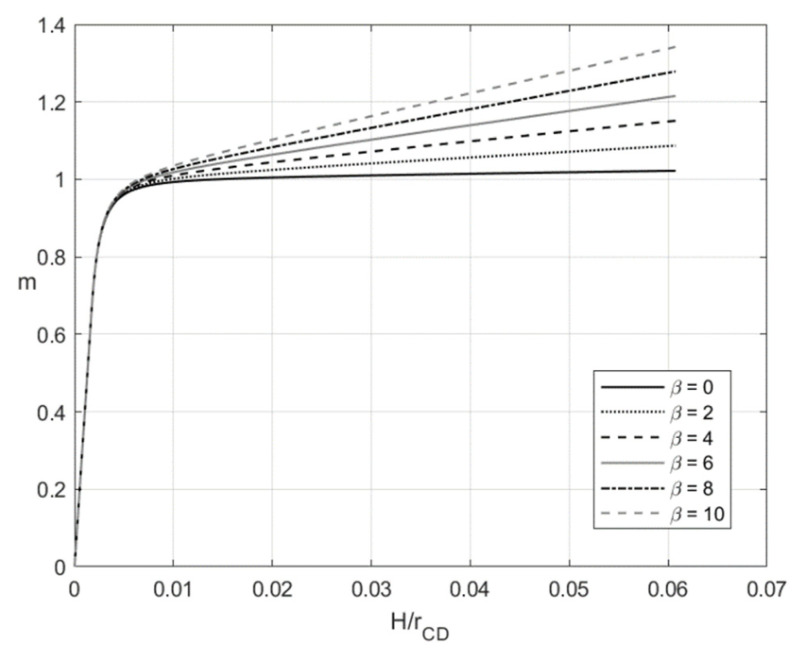
Variation of the dimensionless bending moment with $H/r_{CD}$ at f=0 and several values of β.

**Figure 4 materials-14-01166-f004:**
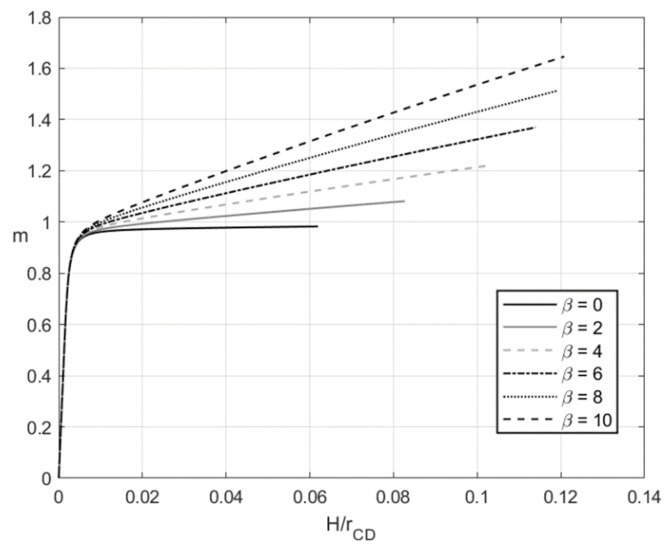
Variation of the dimensionless bending moment with $H/r_{CD}$ at f=0.2 and several values of β.

**Figure 5 materials-14-01166-f005:**
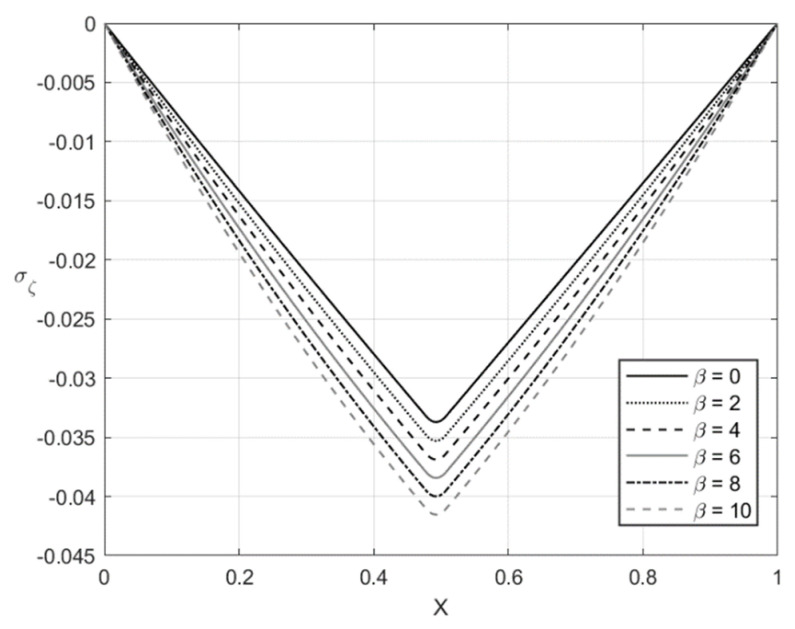
Through-thickness distribution of the stress $\sigma_{\zeta }$ at $H/r_{CD}=0.06$, f=0, and several values of β.

**Figure 6 materials-14-01166-f006:**
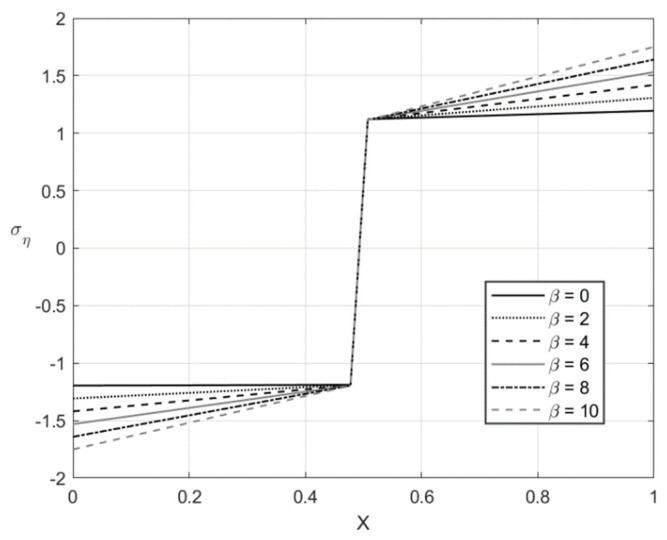
Through-thickness distribution of the stress $\sigma_{\eta }$ at $H/r_{CD}=0.06$, f=0, and several values of β.

**Figure 7 materials-14-01166-f007:**
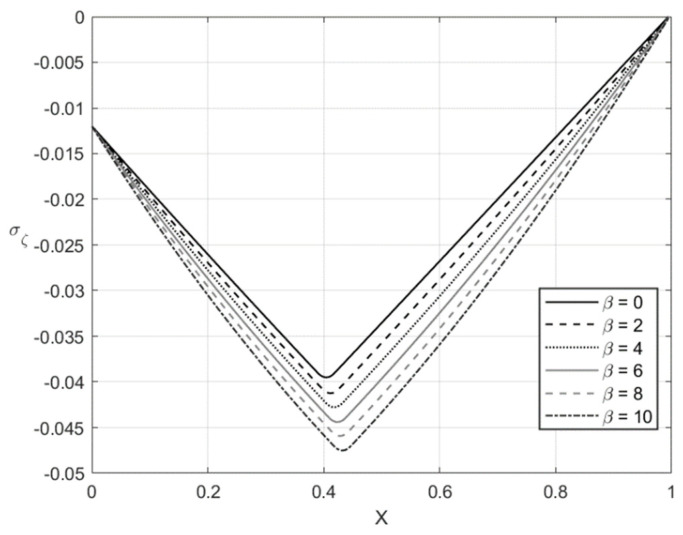
Through-thickness distribution of the stress $\sigma_{\zeta }$ at $H/r_{CD}=0.06$, f=0.2, and several values of β.

**Figure 8 materials-14-01166-f008:**
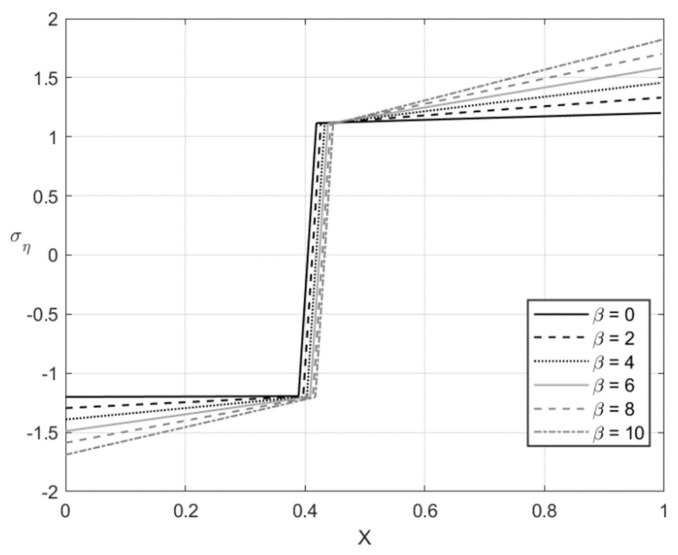
Through-thickness distribution of the stress $\sigma_{\eta }$ at $H/r_{CD}=0.06$, f=0.2, and several values of β.

**Figure 9 materials-14-01166-f009:**
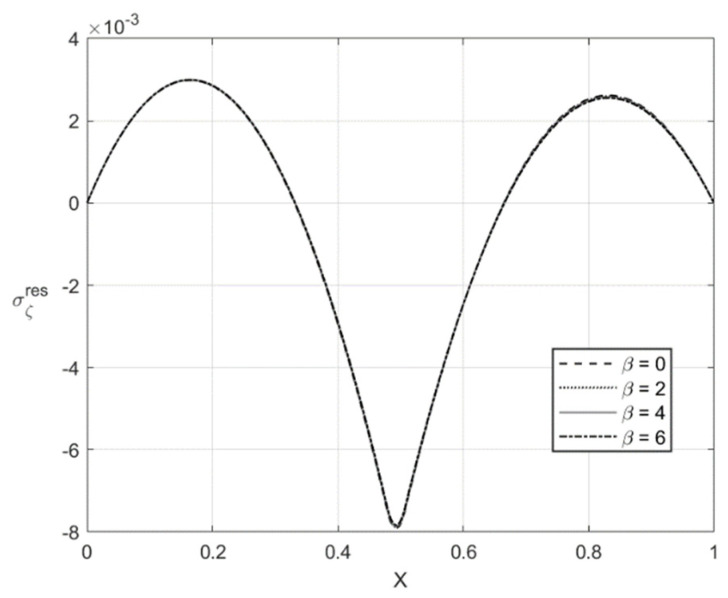
Through-thickness distribution of the residual stress $\sigma_{\zeta }^{res}$ at $H/r_{CD}=0.06$, f=0, and several values of β.

**Figure 10 materials-14-01166-f010:**
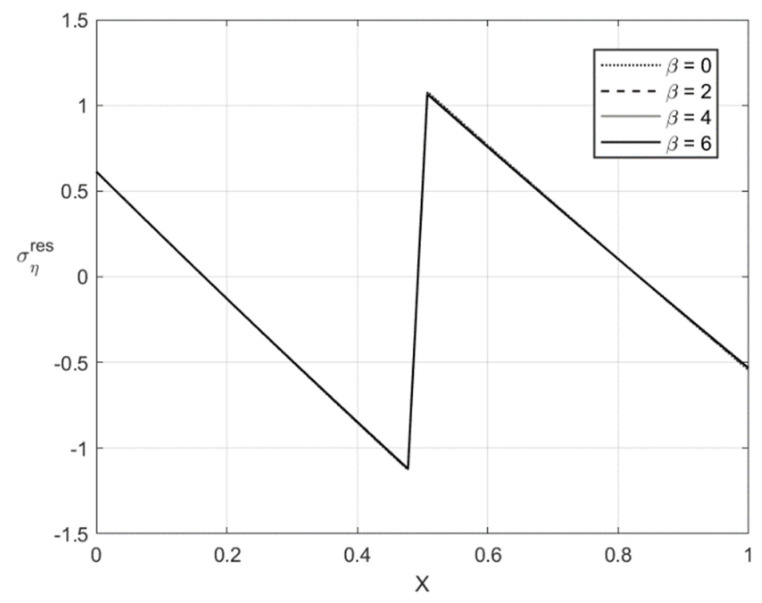
Through-thickness distribution of the residual stress $\sigma_{\eta }^{res}$ at $H/r_{CD}=0.06$, f=0, and several values of β.

**Figure 11 materials-14-01166-f011:**
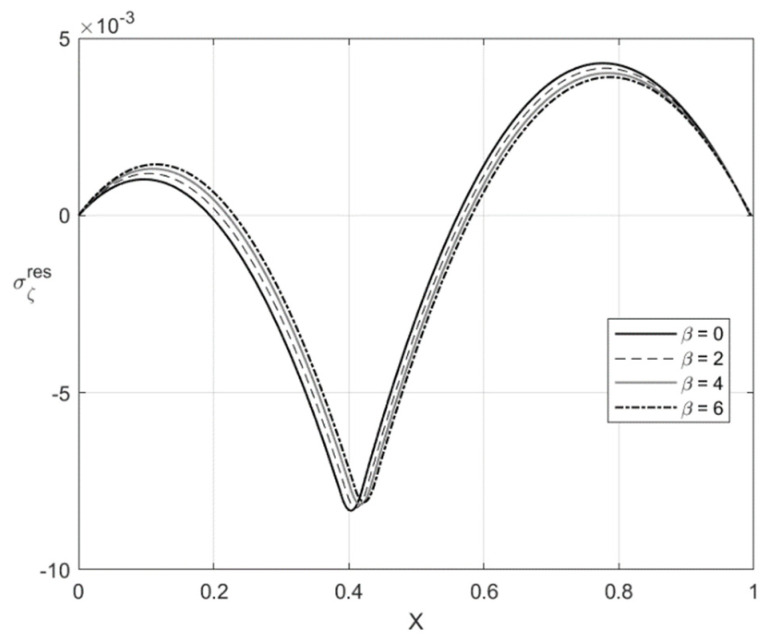
Through-thickness distribution of the residual stress $\sigma_{\zeta }^{res}$ at $H/r_{CD}=0.06$, f=0.2, and several values of β.

**Figure 12 materials-14-01166-f012:**
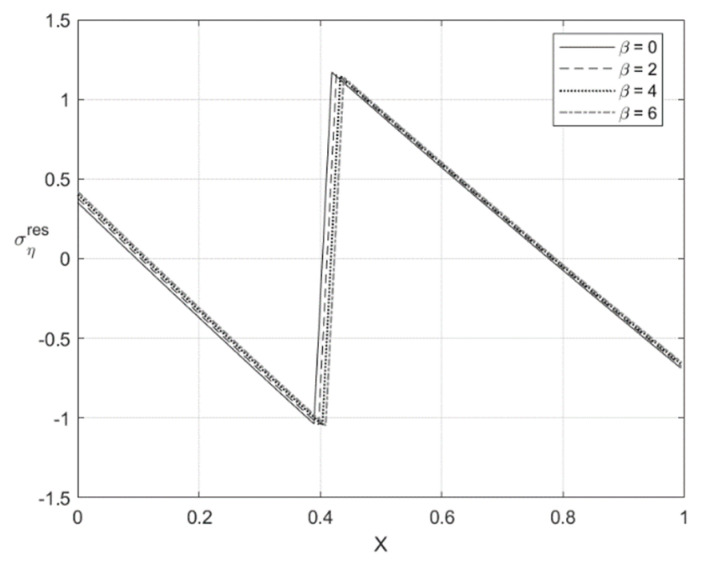
Through-thickness distribution of the residual stress $\sigma_{\eta }^{res}$ at $H/r_{CD}=0.06$, f=0.2, and several values of β.

**Figure 13 materials-14-01166-f013:**
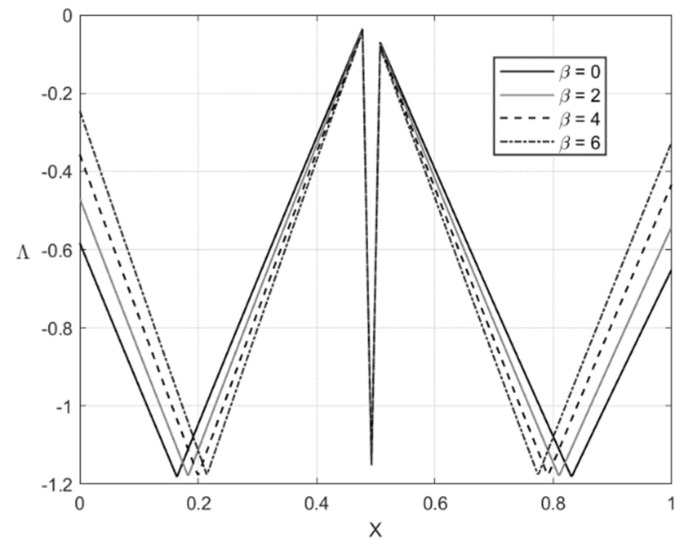
Geometric illustration that the condition (64) is satisfied if f=0 and 0≤β≤6.

**Figure 14 materials-14-01166-f014:**
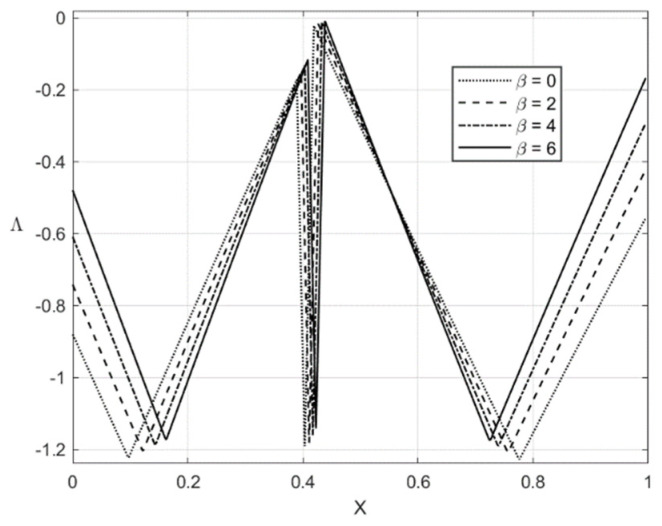
Geometric illustration that the condition (64) is satisfied if f=0.2 and 0≤β≤6.

## Data Availability

Not applicable.

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
