# Peer review of "Finite Plane Strain Bending under Tension of Isotropic and Kinematic Hardening Sheets"

_materials, 2021, doi:10.3390/ma14051166_

Round 1

Reviewer 1 Report

Journal: Materials

Manuscript ID: materials-1111256

Title: Finite Plane Strain Bending Under Tension of Isotropic and Kinematic Hardening Sheets

The authors present a semi-analytical solution in the case of in the plane bending under tension of an elastoplastic sheet with isotopic and kinematic hardening.

This is an extension in the case of these hardenings of an article by one of the co-authors (Reference 17).

It will be necessary to better specify the contributions of this paper in relation to the reference [17].

The conditions at the travel and stress limits need to be better specified. In particular, the deformed shaope after bending is circular (Fig. 1). Is this geometry imposed. How is it justified?

It would be good to show the plastic zones obtained.

Author Response

  1. This is an extension in the case of these hardenings of an article by one of the co-authors (Reference 17). It will be necessary to better specify the contributions of this paper in relation to the reference [17].

Answer: The kinematics equations are not dependent on the constitutive equations, but the stress equations are. We have added Appendix A to clarify the general methodology.

  1. The conditions at the travel and stress limits need to be better specified. In particular, the deformed shape after bending is circular (Fig. 1). Is this geometry imposed. How is it justified?

Answer: There is no mathematical restriction on the kinematics equations. In particular, the radius of the inside surface may tend to zero. The restrictions imposed by the constitutive equations are specified in (60). In the revised manuscript, we have emphasized that the analysis of the process beyond the instant when one of the conditions in (60) is violated is possible but requires additional constitutive equations.

It is an immediate consequence of equation (9) that AB and CD are circular arcs after any amount of deformation.

  1. It would be good to show the plastic zones obtained.

Answer: Figure 2 illustrates the thickness of each plastic region. In the revised manuscript, we have emphasized this meaning of the figure.

Reviewer 2 Report

The manuscript is well written. However, dealing with isotropic and kinematic hardening of plastic materials (in particular steel) has been a hot topic about 20 years ago. Here, I refer to the works by Otto Bruhns /Ruhr University Bochum and his team, who performed both, experimental and theoretical, research on this topic (see e.g. Int. J. Plasticity 1992, Vol8, 331-359). Details on application of isotropic and kinematic hardening were also dealt with in Eur. J. Mech. A, Solids, 1995, Vol 14, 605-632. Of course, well established Cont. Mech. Software can also handle this problem perfectly (e.g. ABAQUS).

Consequently, I cannot find any novelty of this manuscript and cannot recommend this paper for publication from the novelty point of view. However, if the Editor accepts well written papers, which report results by using already established concepts (methods, algorithms etc), then this paper can be published as nice and carefully prepared contribution.

Author Response

Please find the file attached

Reviewer 3 Report

This paper describes a semi-analytical method when a 2D sheet is subjected to mixed loading (tension+bending). An elasto-plastic material model with isotropic and kinematic hardening is used to model the mechanical behaviour of the sheet. The intermediate equations obtained for the plane strain problem are solved numerically. Such semi-analytical models can be used for verification of computational models.

The paper is in an acceptable form. I have two suggestions:

  1. Could the authors include some details on the numerical schemes followed in the paper to solve the integral equations and which programming platform is used to implement the models ?
  2. Could the authors re-define the constitutive models fundamentally in a thermodynamically consistent formalism? i.e., defining the free energy and dissipation function for the elasto-plastic constitutive model used. Based on this formalism could the authors describe the restrictions on the constitutive parameters of the model? The choice is up to the authors if they could follow up this suggestion.

Author Response

The paper is in an acceptable form. I have two suggestions:

1. Could the authors include some details on the numerical schemes followed in the paper to solve the integral equations and which programming platform is used to implement the models ?

Answer: We think that it is a misunderstanding. Our paper contains no integral equations. It is only necessary to evaluate ordinary integrals whose integrands are known functions of their arguments. We use Wolfram Mathematica (https://www.wolfram.com/) to evaluate these integrals. The command NIntegrate solves the problem. The command FindRoot finds solutions of transcendental equations. We have clarified it in the revised manuscript.

2. Could the authors re-define the constitutive models fundamentally in a thermodynamically consistent formalism? i.e., defining the free energy and dissipation function for the elasto-plastic constitutive model used. Based on this formalism could the authors describe the restrictions on the constitutive parameters of the model? The choice is up to the authors if they could follow up this suggestion.

Answer: We do not propose any new material model. We adopt a widely used model from the literature and refer to [21], where a comprehensive discussion of the model has been provided.

Round 2

Reviewer 1 Report

The authors revised their manuscript by addressing the issues which the current review raised. 

Reviewer 2 Report

You can publish it as it is

This manuscript is a resubmission of an earlier submission. The following is a list of the peer review reports and author responses from that submission.

Round 1

Reviewer 1 Report

The revised study remains very similar to the prior works of Alexandrov with kinematic hardening an incremental improvement upon the framework. The paper is better suited for publishing in applied mechanics journals where the framework has been published previously than in Metals.

The authors are encouraged to attempt a finite-element simulation with a tool under an angular stretch bend condition to better appreciate the mechanics and complexities due to contact pressure and the boundary conditions to revise their framework in future work. 

Some responses to the comments are below: 

Comment #2:

A second concern is that the physics and solution of the problem are not validated or appear to be inspired from experiments or simulations. For example, is it possible to experimentally control the stress and bending moments as the paper assumes to generate predictions that provide insight into actual processes? The modelling choices adopted in the manuscript should include some discussion on their suitability and applicability to experiment or practical forming operations where the data can be applied. An angular stretch bend finite-element simulation could provide a benchmark solution to compare against for a typical case of bending under tension found in practice.

Our level of expertise in experimental research is not sufficient to discuss details of measuring or controlling the stress and bending moment. We start our paper with the statement that bending is used as a test for identifying material properties and provided several references. It is impossible to identify these properties without measurements. We believe that the authors of those experimental papers know how to do it. Our solution is exact for the material model and boundary conditions chosen. The only verification that may apply to exact solutions is that they contain no error in derivations. If an exact solution disagrees with experimental results, then the material model is not appropriate, or the boundary conditions are not appropriate, or the experimental results are not accurate. In the case of any of these reasons, it is impossible to improve the theoretical solution for its better comparison with the experimental results. This statement is correct if the material model is completely specified. An advantage of our solution is that the function Fi is arbitrary. For this reason, the solution can be used for identifying the constitutive equations. We have mentioned it in Section 7 of the revised paper. As to finite element simulation, we do not agree. Why do you think that a finite element solution is more reliable than an exact mathematical solution? The latter can be verified without following derivations. One uses the final result, puts it in Wolfram Mathematica (or another similar computing system), and verifies that this final result satisfies the original equations. Exact solutions usually serve as benchmark solutions for verifying numerical codes. We have discussed this issue in Section 7 of the revised manuscript and added two references, [22] and [23].

As mentioned by the Authors in the next comment, the present paper allegedly intends to emphasize the applied aspects of their methodology. However, the paper in its current form does not include any application to a physical forming operation in order to verify the correct implementation of the methodology. The authors are encouraged to attempt to enforce the boundary conditions imposed in the solution in FEA, which would be challenging, and then to recognize that it does not seem possible to accomplish experimentally.

It is the Reviewer’s opinion that the industrially-focused researchers among the audience of the metals journal will not be convinced of the applicability of the proposed methodology even though it was rigorously derived in the prior works of Alexandrov.

The Reviewer does not agree with the statement added on Line 376 that this methodology can be used for verifying numerical codes. The boundary conditions used in the analytical solution will be very complex to implement in a sheet metal bending operation to maintain the load ratios.  

Comment #3:

It is the reviewer's opinion that the manuscript may be better suited to a journal devoted to applied mechanics or applied mathematics rather than the audience of Metals.

We agree with you that the general methodology is more appropriate for a journal devoted to applied mechanics, and it has been published in AAM. The present paper emphasizes the applied aspect (how to apply the general methodology to the material model chosen). The solution is cumbersome, but it does not require any math knowledge beyond a standard course on calculus. Moreover, the basic assumptions are similar to or even less restrictive than those accepted in papers published in journals devoted to metal forming. We have added two references, [20] and [21], to justify this statement.

If the Authors are referring to line 55-59 on page 2 of the revised manuscript, it seems that references were made to sources [18,19] instead. The Reviewer would appreciate it if the Authors could please be more specific in their answer as to what sections in the revised manuscript they are referring to. The Reviewer believes that there is an issue with referencing. Line 61 in the revised manuscript refers to Prager’s law as reference [20] whereas the list of references declares reference [20] as the study of Parsa and Ahkami (2008).

The method consists in assuming transformation equations that connect Eulerian and Lagrangian coordinates, and showing that these equations describe the geometry of plane strain 56 bending at large strains. These equations can be used in conjunction with many constitutive equations 57 [16, 17]. The basic geometric assumptions used in [15] are similar to or even less restrictive than those 58 accepted in papers published in journals devoted to metal forming [18, 19].

Comment #4:

127: The authors state that some pressure P is required to satisfy the equilibrium equation (18). Please interpret this result from a more physical perspective. Is this pressure caused by the stretching force or can this be seen equivalent to a tool contact stress? The presence of tool contact has not been specifically discussed but is known to have a marked influence on the through-thickness stress-strain evolution in most sheet metal forming processes involving bending under tension.

We see no difference for the boundary value problem. One cannot apply the stretching force without applying P. Appendix C shows how to connect P and F. Of course, there should be a tool to support P. We have mentioned the existence of the tool in the revised manuscript.

Referring to Appendix C, computation of the contact pressure was obtained from force equilibrium considering that the resulting force in the tangential direction has to balance the force caused by tool contact pressure. In this case, for pure bending in the absence of tool contact, the force F would be zero since no tool contact stresses prevail. However, in the experience of the Reviewer, this is not the case since the shift in the neutral layer will cause a tangential stress and thus force in the tangential direction which is not zero. The magnitude of the contact pressure could easily be estimated by means of FEA to evaluate the analytical modelling assumptions. 

In short, the tool contact pressure is separate from what one can approximate from the analytical force equilibrium performed. For example, the Hertzian contact stresses would be present starting at the inner layer and are a function of local curvature. These stresses can be very significant and are not captured in the modelling that estimates the pressure from force equilibrium. This should be noted within the analysis as an assumption as the contact stresses affect the through-thickness stress gradient which in turn affects all other aspects of the solution. 

Comment #5:

296: Figure 7 illustrates the shift of the neutral layer towards the concave sheet side for f = 0.2 and different values for beta. Is there a limiting case for which the stretching force is large enough to shift the neutral layer to the concave side or even outside of the sheet thickness? If so, how can this be treated?

The limiting case is tension with no bending. In this case, there is no neutral line. An alternative interpretation is that this line is located at infinity. If F is very large, then there is one plastic region (except for the very beginning of the process when strains are still small). Equation (34) and the integration in Appendix B are valid in this region. In general, this solution is much simpler than the solution provided in the paper. We have mentioned this structure of the solution in Section 7 of the revised manuscript.

The Reviewer is referring to the case where sufficient stretching causes the entire cross-section to be under tensile deformation. The Authors state in the revised manuscript (line 358-364) that in this case, the distribution of stresses and strains is uniform. The Reviewer partly disagrees with this statement since the mild amount of bending will still cause a stress-strain gradient through the sheet thickness. The severity of the gradient should be dependent upon the location of the unstreched and neutral layer.

Reviewer 2 Report

Two observations should be taken into account in the manuscript:
1) In the abstract and conclusion, it is necessary to highlight the novelty "For the first time, an analytical solution was obtained …»;
2) Include the applied aspect of the solution in the abstract, and add the direction of development of this topic in the conclusions.

Reviewer 3 Report

I have checked the re-submitted paper (which I presume is considered now a new submission). There have been some additions but these do not change the character of this work. Unfortunately, the same issues stand - as per my previous review. Thus, I will have to recommend a rejection.